# Arabidopsis assemble distinct root-associated microbiomes through the synthesis of an array of defense metabolites

Enoch Narh Kudjordjie, Rumakanta Sapkota¤, Mogens Nicolaisen ●*

Faculty of Technical Sciences, Department of Agroecology, Aarhus University, Slagelse, Denmark

¤ Current address: Faculty of Technical Sciences, Department of Environmental Science, Aarhus University, Roskilde, Denmark
* mn@agro.au.dk

**Data Availability Statement:** The MiSeq paired end reads for bacterial 16 s rRNA gene (V3-V4) and fungal ITS2 regions have been deposited in the

## Abstract

Plant associated microbiomes are known to confer fitness advantages to the host. Understanding how plant factors including biochemical traits influence host associated microbiome assembly could facilitate the development of microbiome-mediated solutions for sustainable plant production. Here, we examined microbial community structures of a set of well-characterized *Arabidopsis thaliana* mutants disrupted in metabolic pathways for the production of glucosinolates, flavonoids, or a number of defense signalling molecules. *A. thaliana* lines were grown in a natural soil and maintained under greenhouse conditions for 4 weeks before collection of roots for bacterial and fungal community profiling. We found distinct relative abundances and diversities of bacterial and fungal communities assembled in the individual *A. thaliana* mutants compared to their parental lines. Bacterial and fungal genera were mostly enriched than depleted in secondary metabolite and defense signaling mutants, except for flavonoid mutations on fungi communities. Bacterial genera *Azospirillum* and *Flavobacterium* were significantly enriched in most of the glucosinolate, flavonoid and signalling mutants while the fungal taxa *Sporobolomyces* and *Emericellopsis* were enriched in several glucosinolates and signalling mutants. Whilst the present study revealed marked differences in microbiomes of Arabidopsis mutants and their parental lines, it is suggestive that unknown enzymatic and pleiotropic activities of the mutated genes could contribute to the identified host-associated microbiomes. Notwithstanding, this study revealed interesting gene-microbiota links, and thus represents valuable resource data for selecting candidate *A. thaliana* mutants for analyzing the links between host genetics and the associated microbiome.

## Introduction

Plants interact with a vast diversity of microorganisms both above- and belowground, and the outcomes of those interactions may be either beneficial or detrimental to the plant. Essentially, the plant employs a range of strategies such as the action of constitutive and/or induced

NCBI Sequence Read Archive (SRA) database under the accession number PRJNA579829.

**Funding:** Aarhus University (project number 22550) Independent Research Fund Denmark (DFF), grant number 6111-00065B.

**Competing interests:** The authors declare that they have no conflict of interest.

chemical compounds in combination with the plant innate immune system to assemble its associated microbiota [1]. The plant secondary metabolites glucosinolates (GLS) and flavonoids (FLVs) have been widely studied for several microbiota-mediating and plant protective functions [2]. For instance, GLS from the roots of *Brassica* species were found to inhibit microbial pathogens including *Pseudomonas syringae*, *Alternaria brassicicola*, *Gaeumannomyces graminis*, *Botrytis cinerea*, *Fusarium oxysporum* and *Hyaloperonospora parasitica* [3, 4]. The FLVs are well known for their chemoattractant and signalling function in legume-rhizobia interactions resulting in N-fixation and role in plant-mycorrhizal associations, but also as phytoanticipins [5, 6]. Phytohormones serve as signalling molecules in regulating the innate immune network, and salicylic acid (SA), jasmonic acid (JA), ethylene (ET) and abscisic acid (ABA) act as molecular switches in stimulating inducible defense against biotic and abiotic stresses [7, 8]. Owing to its robust and overarching activation of defense repertoires, the immune system is perceived to affect microbial community structures [1, 9, 10].

The biosynthetic pathways and genes involved in GLS [11, 12], FLV [13–15] and defense signalling [16–18] are well described, and research is directed towards exploiting these pathways to study the links between plant gene functions and microbiome assemblage. Several well-characterized mutants of the model plant, *Arabidopsis thaliana* (hereafter Arabidopsis) have become quintessential for studying those relationships. For example, Badri *et al.* [19] reported effects on microbial communities of a mutation in a plant ATP transporter involved in exudation of plant secondary metabolites, and further concluded that individual plant genes are actively involved in the interaction with microbial communities. By using GLS [20], FLV [19] and benzoxazinoid (BX) mutants [21], the influence of plant defensive secondary metabolites on the plant-associated microbiota has been demonstrated. For example, distinct microbiomes were observed in maize parental lines and their isogenic mutants (*bx1*, *bx2* and *bx6*) carrying disruptions in genes encoding enzymes in different steps of the BX pathway [21]. This study further demonstrated a gatekeeper role of BXs in modulating plant-associated microbiomes associated with plant roots. In other studies, the coumarin-impaired mutants, *myb72-2* and *f6'h1* were used to demonstrate the impact of coumarins on microbial community structures [22, 23]. In addition, studies have used Arabidopsis mutants to examine the influence of phytohormones including fatty acid desaturases (FAD) on microbial community structures [24].

Mechanistic processes at the rhizoplane, including the gating role of plant secondary metabolites and defense signalling molecules (DSMs) could control the assembly of host specific microbiomes. We hypothesized that mutations in pathways for the synthesis of certain secondary metabolites and DSMs disrupt the ability of the plant to assemble an optimal microbiome. Findings from other studies using different experimental systems including either a single or a few mutants have reported contrasting effects of plant metabolites on the plant associated microbial community structures. Moreover, previous studies have relied on in-vitro systems where metabolites were exogenously applied and their effect on microorganisms examined. However, such studies do not always reveal the precise effects of these metabolites in a natural system. Our objective in the present study was to assess root microbiome assembly in natural soils in a range of plant mutants with gene disruption in different steps of defense-related biosynthetic or signalling pathways. For this, we selected a range of well-characterized *Arabidopsis* mutants disrupted in GLS, FLV and DSM synthesis and examined their effects on bacterial and fungal communities in a field soil. The analysis of these mutants using identical soils and growth conditions will provide comparable insights of the effects of these mutations on bacterial as well as fungal community structures, the latter has received little attention in previous studies.

## Materials and methods

### Plant material

We used 21 Arabidopsis mutants and their genetic background lines Col_0 and Ler_0 (Fig 1A–1C; **S1 Table in** S1 File). The parental line Col-0 is a natural accession that is maintained as a homozygous line, while Ler-0 carries X-ray induced mutations in the ERECTA gene [25], resulting in distinct chemical profiles [26], root morphology [27, 28], and resistance against *F. oxysporum* [29]. All GLS mutants are in a Col-0 background and similarly for the DSM mutants except for *aba3_2*, which was derived from Ler-0. The flavonoid tt mutants are in a Ler-0 background while pap1_D has Col-0 as its parental line. The GLS (*cyp79B2* and *cyp79B3*), FLVs (*tt3*, *tt5*) and jasmonic acid (*dde2*) mutants were kindly provided by Profs. Judith Bender (Brown University), Wendy Peer (University of Maryland) and Paul Staswich (University of Nebraska), respectively. Other Arabidopsis lines were supplied by the Nottingham Arabidopsis Stock Centre (NASC), UK.

### Experimental design

Arabidopsis seeds were sown in pots (8cm x 8cm x 6cm) with moistened field soil (fine sand 32.2%, coarse sand 52.8%, humus 4.7%, clay 3%, silt 7.3%) [30], pH 5.95, collected from a fallow field at the Jyndevad Research station (N 46˚ 28' 20.716"/E 9˚ 28' 45.347") in Denmark. Microbiome analysis of the soil revealed high abundance of bacterial phyla Acidobacteria and Proteobacteria and the fungal classes Sordariomycetes and Mortierellomycetes [21]. Each pot represented a biological replicate of individual genotypes and was replicated 5 times for all genotypes. Seeds were stratified and pots completely randomized and maintained in a greenhouse under 2017 summer conditions (**S1 Fig in** S1 File). Seedlings (five plants per pot) were maintained by capillary watering (100 ml) and weed removal every week. Sampling was done after 4 weeks of plant growth, by uprooting each plant and gently shaking roots to remove loosely adhering soils. Roots (with remaining attached fine soils) of the 5 plants in each pot was pooled and placed into 2 ml collection tubes, to represent one replicate. The samples were frozen in liquid nitrogen and stored at -20˚C. Subsequently, the samples were lyophilized and ground with 3 sterile metal balls (size 2.88mm) using a Geno/Grinder 2010 at a rate of 1500 rpm for 2 minutes before DNA extraction.

### Sample processing, sequence analysis and statistics

Sample DNA extraction and library preparation were essentially performed as previously described [21]. Briefly, we extracted DNA using the PowerLyzer™ Power Soil® DNA Isolation Kit (Mo Bio Laboratories, Carlsbad, CA, USA). The bacterial primers S-D-Bact-0341-b-S-17, 5′–CCTACGGGNGGCWGCAG–3′ and S-D-Bact-0785-a-A-21, 5′–GACTACHVGGGTATC TAATCC–3′ [31] and the fungal primers fITS7, 5′–GTGARTCATCGAATCTTTG–3′ and ITS4, 5′–TCCTCCGCTTATTGATATGC–3′ [32] were used to amplify the V3/V4 region of the bacterial 16S rRNA and the fungal internal transcribed spacer 2 (ITS2) region, respectively. A dual indexing strategy was used, and PCR conditions were as described [21]. For detailed library and PCR conditions see **Additional file 1 in** S1 File. Samples were sequenced using the Illumina MiSeq platform at Eurofins MWG (Ebersberg, Germany). All sequence files were deposited in the NCBI Sequence Read Archive (SRA) under the accession number PRJNA579829.

Sequence analysis including demultiplexing, operational taxonomic units (OTUs) clustering at 97% similarity cutoff value, chimera detection and removal, and OTU table creation were performed using VSEARCH version 2.6 [33], as described in [21]. Taxonomy assignments were carried out in QIIME version 1.9 [34], respectively using SILVA 132 [35] and

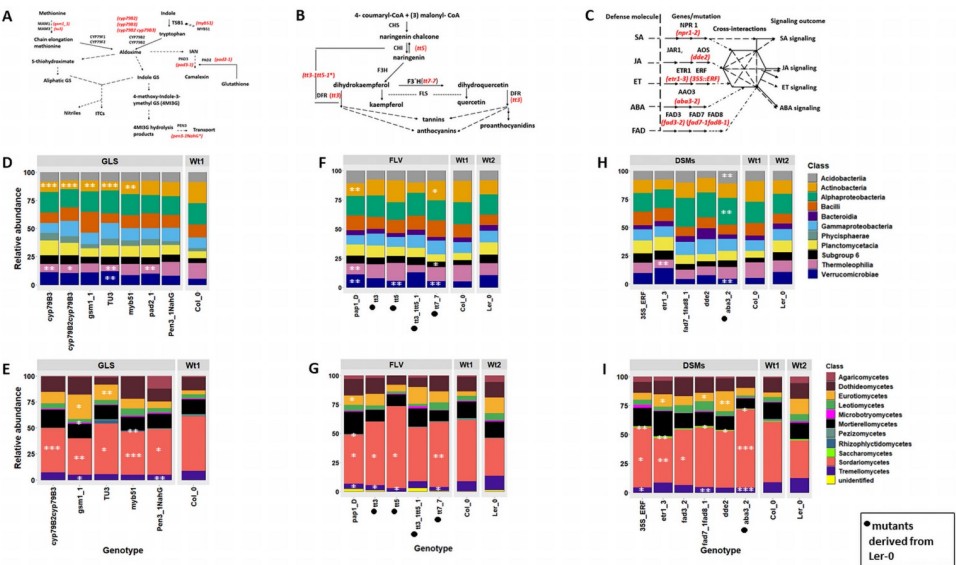

**Fig 1. Plant biosynthetic and defense signalling pathways and microbial relative abundances. A**) Biosynthetic pathway of aliphatic and indolic GLS in Arabidopsis. Adapted from Frerigmann et al, (2014). IAN (Indole-3-acetonitrile), TSB1 (tryptophan synthase beta subunit 1). **B**) The flavonoid biosynthetic pathway. CHI, chalcone isomerase; CHS, chalcone synthase; DFR, dihydroflavonol reductase; F3H, flavonol 3-hydroxylase; FLS, flavonol synthase. *(double mutant). Adapted from Buer et al., (2009, 2010). **C**) The defense signalling pathways. The phytohormones salicylic acid (SA), jasmonic acid (JA), ethylene (ET), abscisic acid (ABA) mediate defense signalling in plants. Fatty acid desaturase (FAD) is also involved in defense including the regulation of JA and SA pathways. Defense signal interactions to fine-tune defense signalling outcomes is also shown. NPR1 (Non-expressor of PR genes1), AAO3 (Abscisic acid synthase), AOS (Allene oxide synthase). Both disrupted genes from which mutants were derived and mutants (indicated in red) are shown. Class-level relative abundances (RAs) of microbial communities observed in Arabidopsis genotypes. **D**) Bacterial and **E**) fungal RAs at class level in GLS mutants and the parental line, Col_0. **F**) Bacterial and **G**) fungal RA at class level in FLV mutants and respective parental lines, Col_0 and Ler-0. **H**) Bacterial and **I**) fungal RAs at class level in DSMs and their respective parental lines, Col_0 and Ler-0. Test of statistical significance was performed by comparing each mutant to their respective background parental lines, and the significantly affected taxa are indicated as *** (P<0.001), ** (P<0.01) and * (P<0.05). ANOVA tests were followed by the Tukey–Kramer *post hoc* test using the Benjamini and Hochberg (BH) FDR for multiple comparisons. Only the most abundant 10 taxa were used in this analysis. Microbial taxa with small effect sizes were removed by filtering (effect size = 0.8). The analysis was performed using the STAMP software (v2.1.3).

UNITE version 7.2 [36] reference databases for bacterial and fungal OTUs. Downstream data exploration and visualization was performed in the R statistical package [37] using the ggplot2 (v3.3.2) package. Microbial community diversity estimations including alpha and beta diversities, species richness and dissimilarity were performed using the 'vegan' package [38] and 'phyloseq' [39]. We applied a minimum cutoff of 700 and 500 reads respectively, for bacteria and fungi and the reads and OTU distribution were visualized (**S2 Fig in** S1 File). Following this, samples with less than 3 replicates were removed and OTUs represented in less than 3 samples in the total dataset were also excluded prior to downstream analysis.

To determine statistically significant differences in taxonomic profiles, we performed a multiple group analysis by comparing sequences assigned to different class level taxa (top 10) in mutants and their respective parental lines (Col_0 or Ler-0) using the STAMP software v2.1.3 [40, 41]. Genotypes were compared by ANOVA, followed by the Tukey–Kramer *post hoc* test (p < 0.05) using the Benjamini and Hochberg (BH) FDR for multiple comparisons. Class taxa with small effect sizes were removed by filtering (effect size = 0.8).

Bacterial and fungal alpha diversities (observed and Shannon) were estimated by rarifying the OTU table 100 times at a depth of 700 reads for bacteria and 500 reads for fungi and the

mean of the diversity estimates of 100 trials was used. To identify the effect of host genotype on alpha diversity, we partitioned the data for each mutant and parental line and performed Kruskal-Wallis test, followed by pairwise Wilcoxon rank sum tests with Benjamini-Hochberg correction. The OTU tables were transformed to relative abundances (RAs) for beta diversity analysis for each genotype groups (GLS, FLV and DSM mutants), and further visualized using unconstrained principal coordinates analysis (PCoA). Permutational multivariate analysis of variance (PERMANOVA) based on a Bray-Curtis dissimilarity matrix was carried out by using 1000 permutations to detect significant differences in the overall bacterial and fungal community composition, using the "adonis" test from the "vegan" package [38]. We further confirmed significant genotypic variations on microbial community structures using generalized linear models (GLMs), where genotype was fitted to each OTU using the mvabund package (function *manyglm* with a negative binomial distribution) [42]. The likelihood-ratio test was used, and p-values were adjusted for multiple testing. To further analyse specific genotype effects, we split the data for each mutant and parental line and performed PERMANOVA analysis. We performed differential analysis using DESeq2 (version 1.22.2) [43] to determine bacterial and fungal taxa that were significantly different in mutants and parental lines.

## Results

We studied microbiome composition in Arabidopsis plants carrying mutations in specific steps of the glucosinolates (GLS), flavonoids (FLV) and defense signaling molecules (DSM) pathways (Fig 1A–1C). The bacterial community profiling yielded a total of 717,379 sequence reads clustering into 6,471 OTUs while the fungal profiling yielded 703,675 sequences which clustered into 344 OTUs after quality filtering. The sequence read statistics are provided (**S2 Table in** S1 File).

### Microbial abundance is affected in most mutants

Bacterial and fungal class relative abundances were distinct in the GLS, FLV and DSM mutants compared to parental lines (Fig 1). We found the most highly significant differences (P<0.001) in the mean abundance of reads belonging to the bacterial classes Actinobacteria, Thermoleophilia and Verrucomicrobiae in the GLS upstream mutants *cyp79B3*, *cyp79B2cyp79B3* and *tu3* (Fig 1D). Both *cyp79B3 and cyp79B2cyp79B3* have upstream mutations that lacks the ability to form aldoxime from tryptophan and, thus, are deficient in indolic GLS [44, 45], while *tu3* lacks $C_6$, $C_7$, and $C_8$ aliphatic glucosinolates [12]. Similarly, the fungal class Eurotiomycetes was highly enriched in *gsm1_1* and had the least abundance in Col_0 (Fig 1E). Dothidiomycetes was strongly enriched in *myb51*(impaired in IGS and camalexin synthesis) (Fig 1E).

A multiple test analysis revealed significant enrichment of a number of bacterial taxa in FLV mutants *pap1_D (FLV* overexpressed mutant) and *tt7_7* compared to their parental lines. Verrucomicrobiae was strongly enriched in *tt5* and *tt7_7* (deficient in flavonoid 3'-hydroxylase activity and lacking orthodihydroxy flavonoids such as quercetin and kaempferol) compared to Ler_0 (Fig 1F). Similarly, we found significant differences in the relative abundances of Sordariomycetes and Tremellomycetes among the mutants (Fig 1G). In the DSM mutant *aba3_2*, Acidobacteria and Alphaproteobacteria were enriched while Verrucomicrobiae was depleted compared to the parental line Ler_0 (Fig 1H). Thermoleophilia was strongly depleted in *etr1_3*. Sordariomycetes was enriched, while Tremellomycetes was depleted in *aba3_2* (Fig 1I). The relative abundance of other fungal taxa including Eurotiomycetes and Leotiomycetes were also significantly affected.

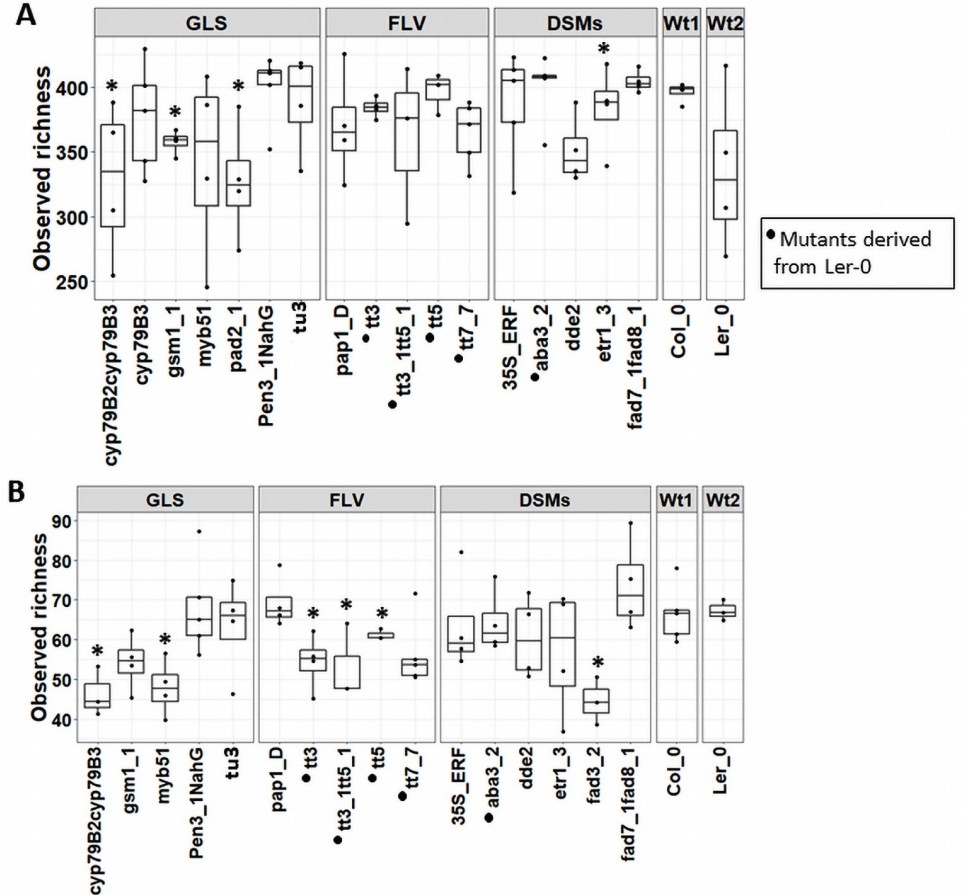

**Fig 2. Alpha diversity (observed) of bacterial (A) and fungal (B) communities in roots of Arabidopsis secondary metabolite and signalling mutants and their parental lines (Col_0 and Ler-0).** Significant differences in alpha diversity in secondary metabolite and signalling mutants and their parental lines are indicated as *, **, and *** for P <0.05, P<0.01 and P<0.001 respectively. The analysis was performed using the Kruskal-Wallis test.

## GLSs, FLVs and DSMs all affect root microbial diversity

We found significant differences in bacterial observed richness (P<0.05) in the GLS mutants *cyp79B2cyp79B3*, *gsm1_1* and *pad2_1* and in the DSM mutant *dde2* (Fig 2, **S3 Table in** S1 File). Shannon diversity was also significantly different (P<0.05) among the *pad2_1* and *dde2* mutants (**S3A Fig in** S1 File, **S3 Table in** S1 File). Bacterial alpha diversity was not significantly different in any of the FLV lines; however, the observed richness was higher in these mutants (the tt lines) compared to Ler_0. For fungi, the observed richness was significantly lower (P< 0.05) in the GLS *cyp79B2cyp79B3*, *myb51*, the FLVs *tt3*, *tt3_1tt5_1*, *tt5* and the DSM mutant *fad3_2* compared to Ler_0 (Fig 2, **S3 Table in** S1 File). In addition, Shannon diversity was significantly different (P<0.05) in the DSM mutant *aba3_2* (**S3B Fig in** S1 File, **S3 Table in** S1 File).

Bacterial beta diversity analysis and visualization using PCoA ordination plots showed a clear separation of GLS mutants from Col_0 (Fig 3A). PERMANOVA revealed significant differences on the bacterial communities (Adonis, bacteria: $R^2$ = 0.31, P< 0.001; Table 1; **S4 Table in** S1 File,). We likewise observed clustering in the fungal PCoA plots, except that Col_0

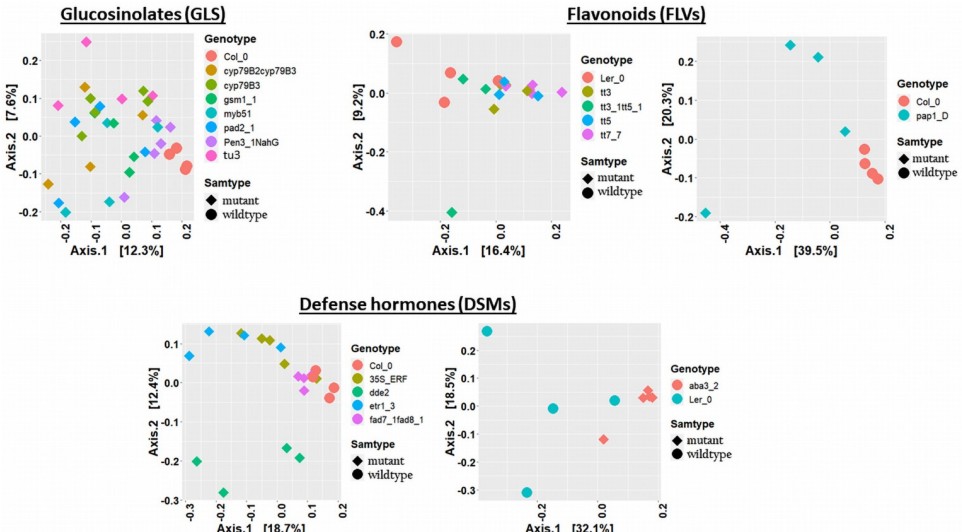

**Fig 3. Principal coordinate analysis (PCoA) plots of bacterial communities in Arabidopsis secondary metabolite and signalling mutants, and their parental lines.** PCoA of bacterial communities in GLS (**A**), FLV (**B**) and DSM (**C**) genotypes. Individual genotypes and sample groups are shown in different colours and shapes.

and *pen3_1_NahG* (impaired in both IGS and SA) clustered together (Fig 4A), and the GLS mutation also significantly affected fungal communities (Adonis, $R^2 = 0.37$, P< 0.001, Table 1; **S5 Table in** S1 File). The *tu3* mutant explained the highest proportion of the variation on both bacterial (Adonis, $R^2 = 0.34$, P<0.01, Table 1) and fungal (Adonis, $R^2 = 0.41$, P<0.01, Table 1) communities. The *pen3_1_NahG* mutant, which is disrupted in both metabolism and transport of tryptophan-derived secondary metabolites and SA synthesis, expectedly had a strong effect on bacterial communities. However, *pen3_1_NahG* line also carries mutation in PEN3 gene known for cell-wall defense enhancement [46], thus, its effect on the microbiome should be interpreted cautiously. Similarly, PCoA ordination plots showed clustering in the FLV mutants and parental lines in both bacterial and fungal datasets (Figs 3B and 4B) and significant differences were confirmed by PERMANOVA for bacterial (Adonis, $R^2 = 0.36$, P< 0.001, Table 1; **S4 Table in** S1 File)) and fungal (Adonis, $R^2 = 0.38$, P<0.01, Table 1; **S5 Table in** S1 File) community structures. Further PERMANOVA analysis using datasets consisting of individual mutants and their parental lines revealed the highest effects of *tt3_1_tt5_1* and *tt7_7* mutants on bacterial (Adonis; $R^2 = 0.28$, P<0.05, Table 1) and fungal (Adonis; $R^2 = 0.33$, P<0.05, Table 1) communities, respectively. A minor but significant effect of *tt3* (Adonis; $R^2 = 0.31$, P<0.05, Table 1) and *pap1_D* (Adonis; $R^2 = 0.33$, P<0.05, Table 1) was detected on the fungal communities.

The DSM mutants and their parental lines likewise showed clear clustering of bacterial and fungal communities (Figs 3C and 4C) and were further confirmed to be significantly different (bacteria: Adonis; $R^2 = 0.39$, P<0.001, Table 1; **S4 Table in** S1 File, and fungi: Adonis; $R^2 = 0.40$, P<0.001, Table 1; **S5 Table in** S1 File). PERMANOVA performed on the individual mutants and their parental lines showed that the highest proportion of variation was contributed by the *dde2* and *etr1_3* mutations on the bacterial communities (*dde2*: Adonis; $R^2 = 0.33$, P<0.001; *etr1_3* Adonis; $R^2 = 0.33$, P<0.001, Table 1), and the *aba3_2* mutation on the fungal communities (Adonis; $R^2 = 0.54$, P<0.001, Table 1).

**Table 1. Permutation analysis of variance (PERMANOVA) for individual mutants and background parental lines.** Adonis tests were based on Bray-Curtis dissimilarity matrices for both bacterial fungal communities using 1000 permutations.

| Dataset/ Factor | Genotype description | Bacteria ($R^2$) | Fungi ($R^2$) |
|---|---|---|---|
| GLS (Mut1) | GLS mutants and parental lines | 0.31*** | 0.37*** |
| FLV (Mut2) data | FLV mutants and parental lines | 0.36*** | 0.38** |
| DSM (Mut3) | Defense signalling mutants and parental lines | 0.39*** | 0.40*** |
| **GLS** | | | |
| Col_0_cyp79B3 | IGS partial disruption) | 0.26** | - - |
| Col_0_cyp79B2cyp79B3 | Lacks IGS and camalexin (Blocked in the production of I3AOx) | 0.28* | 0.33* |
| Col_0_myb51 | IGS synthesis disruption | 0.26* | 0.30* |
| Col_0_gsm1_1 | Reduced amounts of many aliphatic glucosinolates | 0.26* | 0.34* |
| Col_0_tu3 | Deficient in aliphatic GLS with heptyl and octyl core groups | 0.34* | 0.41* |
| Col_0_ pad2-1 | Partially blocks camalexin | 0.28* | - - |
| Col_0_ pen3_1_NahG | Disruption in both IGS synthesis and SA signalling pathways. | 0.20** | 0.16* |
| **FLV** | | | |
| Ler-0_tt7_7 | Deficient in flavonoid 3'-hydroxylase activity and lacks orthodihydroxy flavonoids such as quercetin and kaempferol | 0.23* | 0.33* |
| Ler-0_tt3 | Excess quercetin, kaempferol | - - | 0.31* |
| Ler-0_tt5 | Low-level quercetin production | n.s | n.s |
| Ler-0_ tt3_1tt5_1 | Double mutation, disruption of the synthesis of brown pigment | 0.28* | n.s |
| Col_0_pap1_D | Overexpressed (anthocyanin) mutant | n.s | 0.35* |
| **DSMs** | | | |
| Col_0_dde2 | JA deficient | 0.33* | 0. 41* |
| Ler-0_ aba3_2 | ABA deficient | 0.24* | 0.54* |
| Col_0_ etr1_3 | Ethylene responsive | 0.33* | 0.32* |
| Col_0_35S::ERF | Ethylene (overexpressed) | 0.19* | 0.24* |
| Col_0_ fad3_2 | Fatty acid desaturase (FAD) deficient | - - | 0.25* |
| Col_0_ fad7_1fad8_1 | FAD double mutation | 0.31* | - - |

Significance of test indicated as

*** for $p < 0.001$,

** $p < 0.01$,

*$p < 0.05$ and $R^2$ for the proportion of variation explained. I3AOx (Indole-3-aldoxime) IGS (indole glucosinolate).

¯¯ not computed due to few sample (<3) replicates and n.s (not significant).

## GLS, FLV and DSM mutations specifically enrich or deplete microbial taxa

Next, we performed differential analysis to determine bacterial and fungal OTUs (bOTUs and fOTUs, respectively) that had significantly different relative abundances between mutants and parental lines. Most of the differentially affected genera belong to dominant microbial taxa and are mostly enriched than depleted when secondary metabolite and signaling pathways are altered, except for flavonoid mutations on fungi communities (**S6 Table in** S1 File). For GLS, the highest numbers of differentially abundant bOTUs were observed in upstream mutated lines *tu3* and *cyp79B3* (Fig 5A; **S7 Table in** S1 File). Bacterial phyla Actinobacteria, Alphaproteobacteria and Bacteroidetes were among the most highly enriched taxa in many of the GLS mutants. bOTUs assigned to the genera *Nocardioides* and *Azospirillum* were the most highly enriched in several GLS mutants. Other significantly enriched genera included *Streptomyces* and *Fluviicola* in mutants such as *tu3*, *myb51* and *cyp79B3* (Fig 5A; **S7 Table in** S1 File). Planctomycetes genera were mostly depleted in *cyp79B3*. *Massilia* and *Flavobacterium* were also significantly enriched in *cyp79B3* and *tu3*. These results indicate that both indolic and aliphatic GLSs regulate microbial members.

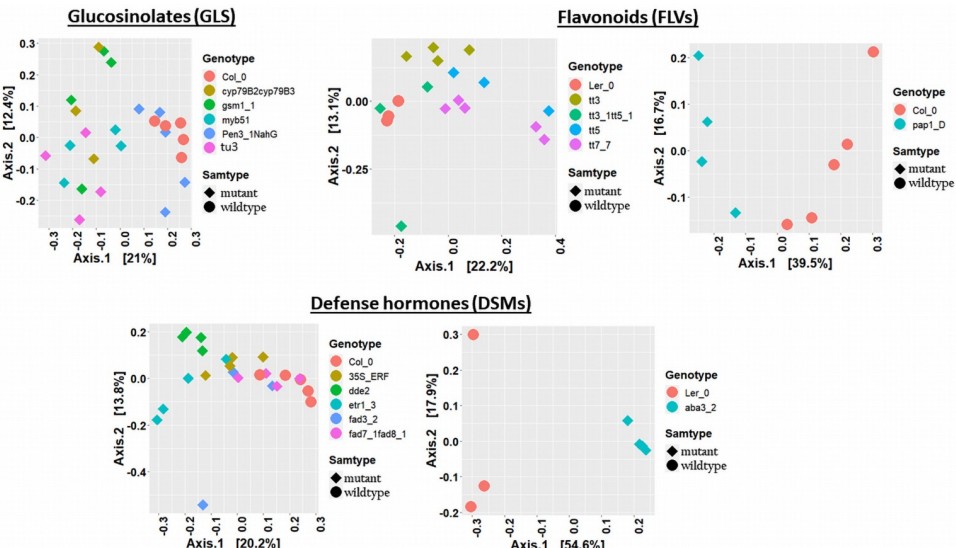

**Fig 4. Principal coordinate analysis (PCoA) plots of fungal communities in Arabidopsis secondary metabolite and signalling mutants, and their parental lines.** PCoA of fungal communities in GLS (**A**), FLV (**B**) and DSM (**C**) genotypes. Individual genotypes and sample groups are shown in different colours and shapes.

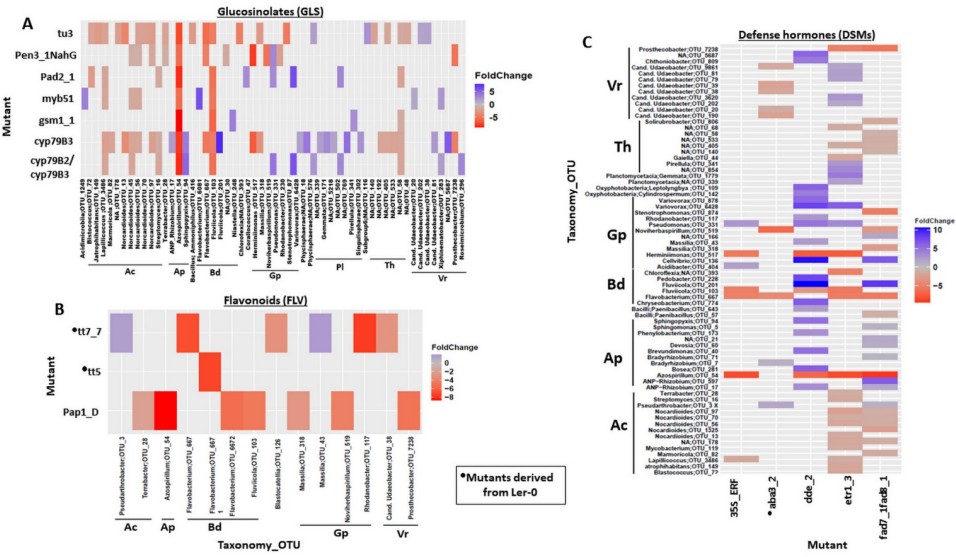

**Fig 5. Heat map of differentially abundant bacterial OTUs identified by comparing secondary metabolite and signalling mutants and their parental lines. A)** Heat map of differentially abundant bacterial OTUs identified by comparing GLS mutants and parental lines. **B)** Heat map of differentially abundant bacterial OTUs identified by comparing FLVs mutants and parental lines. **C)** Heat map of differentially abundant bacterial OTUs identified by comparing DSMs mutants and parental lines. Foldchanges are indicated in blue and red respectively for parental lines and mutants. *ANP-Rhizobium* (*Allorhizobium-Neorhizobium-Pararhizobium-Rhizobium*), *Candidatus Udaeobacter* (*Cand. Udaeobacter*), *Candidatus Xiphinematobacter* (*Cand. Xiphinematobacter*), Ac (Actinobacteria), Ap (Alphaproteobacteria), Bd (Bacteroidetes), Gp (Gammaproteobacteria), Pl (Planctomycetes), Th (Thermoleophilia), Vr (Verrucomicrobiae). Analysis was performed using DESeq2 package.

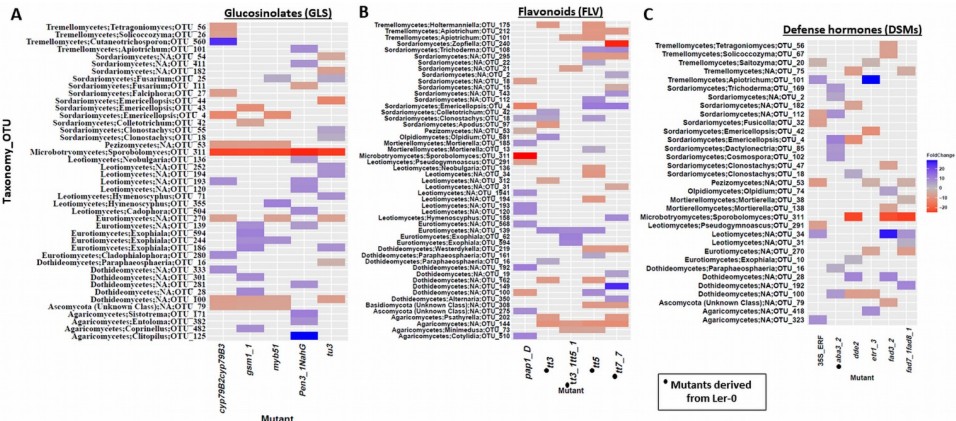

**Fig 6. Heat map of differentially abundant fungal OTUs identified by comparing secondary metabolite and signalling mutants and their parental lines. A)** Heat map of differentially abundant fungal OTUs identified by comparing GLS mutants and parental line. **B)** Heat map of differentially abundant fungal OTUs identified by comparing FLVs mutants and parental line. **C)** Heat map of differentially abundant fungal OTUs identified by comparing DSMs mutants and parental line. Foldchanges are indicated in blue and red respectively in parental lines and mutants. *ANP-Rhizobium* (*Allorhizobium-Neorhizobium-Pararhizobium-Rhizobium*), *Candidatus Udaeobacter* (*Cand. Udaeobacter*), *Candidatus Xiphinematobacter* (*Cand. Xiphinematobacter*). The analysis was performed using DESeq2 package.

We found the lowest number of significantly affected bOTUs in the FLV mutants, with most of these taxa enriched in *tt7_7* and *pap1-D* (Fig 5B; **S8 Table in** S1 File). bOTUs belonging to *Flavobacterium* were enriched in the FLV mutants, and *Rhodanobacter* and *Azospirillum* were significantly enriched in *tt7_7* and *pap1-D*, respectively. The DSM mutant *dde2* was mostly depleted in Proteobacteria and Bacteroidetes genera while Actinobacteria genera were enriched in *etr1_3* and *fad7_1fad8_1*. Other mutants such as *35S::ERF*, *aba3_2* and *dde_2* were enriched in *Azospirillum*, *Flavobacterium* and *Fluviicola* (Fig 5C; **S9 Table in** S1 File).

fOTUs assigned to *Sporobolomyces* and *Emericellopsis* were the most highly enriched genera in the *cyp79B2cyp79B3* and *tu3* mutants (Fig 6A; **S10 Table in** S1 File). In addition, the fungal family Ceratobasidiaceae was strongly enriched in *tt3*, *tt5*, *tt7_7* and *pap1_D*. A blast analysis of reads assigned this taxon to *Waitea circinata/Rhizoctonia spp*. These strong enrichment of the family Ceratobasidiaceae in *tt3*, *tt5*, *tt7_7* and *pap1_D* compared with parental lines could suggest an antagonistic effect of FLVs on fungal members of this taxa. We also observed differential abundances of notable antagonistic fungi such as *Clonostachys rosea* in *tt3*, *tt5* and *pap1_D*, and the genus *Trichoderma* in *tt5* and *tt7_7* (Fig 6B; **S11 Table in** S1 File). Similarly, in the DSM mutants, *Sporobolomyces* was significantly enriched in *dde2*, *fad3_2* and *fad7_1fad8_1*, while *Emericellopsis* was highly enriched in *dde2* and *etr1_3*. Conversely, *aba3_2* was strongly depleted in *Emericellopsis and Trichoderma* while the ethylene mutants *etr1_3* and *35S::ERF* had increased abundances of *Saitozyma* (Fig 6C; **S12 Table in** S1 File).

## Discussion

### GLSs have notable effects on the host root-associated microbiome

Our results revealed notable effects of glucosinolates (GLS) mutants on relative abundances, and alpha- and beta-diversities of microbial communities, suggesting that specific GLS metabolites affect the composition of the root microbiome. The separation of GLS mutants from the parental line in the PCoA plots suggests differential effects of the GLS mutations on the

Arabidopsis microbiome, thus supporting previous studies [47]. Specifically, *tu3* that carries a gene disruption upstream of the aliphatic GLS pathway, had the strongest effect on both bacterial and fungal communities. Aliphatic GLS and their hydrolysis products have been reported to have higher effects on microorganisms compared to indole glucosinolates (iGLS) [48]. The toxicity of aliphatic GLS towards microorganisms is attributed to the complex degradation products isothiocyanates, thiocyanates, oxazolidinethiones and nitriles that are produced from the enzymatic cleavage of glucosinolates by myrosinase [49].

The strong effect of *tu3* also confirmed that gene disruptions at the initial steps of a biosynthetic pathway generally have more pronounced effects on the host-associated microbiota [21]. However, other mutants including *cyp79B3* and the double mutant *cyp79B2cyp79B3* in the same indole GLS pathways or the *gsm1_1* in the aliphatic GLS pathways, with upstream gene disruptions only had minor, but significant, effects on the microbiome. These results suggest a differential regulatory role of metabolic genes and their effects on host associated microbiota. Comparatively, the variations in glucosinolates profiles could cause the differential effects of mutants *tu3* and *gsm1-1* on microbial communities. The *tu3* produce glucosinolates that are deficient in *gsm1-1* (that is, aliphatic glucosinolates with butyl, pentyl, or hexyl core groups) but lacks aliphatic glucosinolates with heptyl and octyl core groups [3, 12]. Brader *et al.* [50] found differences in the induction of the CYP79B2 and CYP79B3 genes upon treatment with culture filtrates of the bacterium *Erwinia carotovora*. Hence, it is possible that unknown enzymatic and pleiotropic activities of the mutated genes could contribute to the observed differences of microbial communities. Furthermore, Ludwig-Müller *et al.* [51] reported that several TU mutants, having different contents of GLS intermediate products, developed varying degrees of clubroot disease symptoms caused by *Plasmodiophora brassicae*. Together, these suggest distinct gene functions in GLS pathways which possibly underline mechanistic processes in microbiome assembly.

GLSs had distinct effects on specific microbial groups as confirmed by the identification of a range of bacterial and fungal taxa responding to the different GLS mutations. The increased abundance of individual bacterial genera such as *Azospirillum* and *Fluviicola* as well as the fungal genera *Sporobolomyces* and *Emericellopsis* in several of the GLS mutants further confirmed the selective effects of metabolites on individual microbial taxa and thus corroborates previous studies [21, 23]. For instance, soils amended with isothiocyanates (allyl, butyl, phenyl, and benzyl ITC) were reported to affect fungal communities more dramatically than bacterial communities [52]. The authors observed changes in community composition including increased *Humicola* abundance in allyl ITC and *Mortierella* abundance in butyl ITC amended soils, while the bacterial phylum Firmicutes temporally increased in response to amendment with allyl ITC [52]. Plant metabolic compounds with antimicrobial properties including GLS are known to be part of the boundary layers of the root rhizoplane that modulate root microbiome assemblage [53]. Other genera such as *Nocardioides* and the plant beneficial taxa *Streptomyces* and *Flavobacterium* were also enriched in most of the GLS mutants. *Azospirillum* contains several beneficial species, widely known for their plant growth promoting traits including nitrogen fixation and synthesis of phytohormones and other compounds required for both biotic and abiotic stress tolerance [54]. The yeast *Sporobolomyces*, that was enriched in the GLS lines and the other mutants, is an abundant member of the plant mycobiome [55, 56] and is antagonistic against pathogens [57]. Also, the strongly enriched genus *Emericellopsis* in both the indole GLS mutants *cyp79B2cyp79B3*, *myb51* and aliphatic GLS lines *gsm1_1* and *tu3* is known to possess biocontrol traits via the antimicrobial compound emericellipsin A, and have been shown to suppress the pathogen *Aspergillus niger* [58]. In addition, the strong enrichment of *Fusarium* in *pen3_1_NahG* could suggest that both physical and chemical barriers (GLS and SA) affect this genus [59, 60].

## Flavonoids have a higher effect on root-associated fungal communities

Flavonoids (FLVs) are some of the most studied phytochemicals due to their profound role in plant-microbe interactions. Analysis of microbiota from FLV mutants impaired in different steps of the FLV pathway revealed differential effects on bacterial and fungal communities. While the FLV mutations did not affect bacterial alpha diversities significantly, weak but significant effects of the individual FLV mutations were observed on microbial community composition. The FLV mutant *tt7_7* which lacks orthodihydroxy flavonoids (for example quercetin, naringenin, genistein, luteolin, daidzein, and morin), and accumulates pelargonidin rather than the cyanidin found in wild-type plants [61], had significant effects on both bacterial and fungal communities. Differential responses of pelargonidin and cyanidin to both fungi and bacteria have been reported earlier [62]. The orthodihydroxy flavonoids have been shown to mediate plant-microbe interactions, especially in nodule formation and in enhancing arbuscular mycorrhizal colonization, and by inhibiting bacterial and fungal pathogens [63, 64]. Hence, the disruption of the synthesis of these FLVs would likely affect microbial communities and the strong enrichment of the bacterial genera *Flavobacterium* and *Rhodanobacter* in *tt7_7* could suggest a modulating role of orthodihydroxy flavonoids on bacterial communities. Similarly, the *pap1_D* which accumulates anthocyanin pigments, (mainly cyanins) and other secondary metabolites [65, 66] significantly affected only the fungal communities.

The *tt3* mutant which accumulates high concentrations of both kaempferol and quercetin [13] had a more profound effect on fungal communities compared to the bacterial communities. Both kaempferol and quercetin are highly secreted in Arabidopsis [67], and their accumulation were likely to affect microbial communities. Quercetin enhances mycorrhizal-plant symbiosis by stimulating host penetration and hyphal growth [5], while kaempferol inhibits germination of pathogenic fungal spores [68]. However, Vikram *et al.* [69] reported that kaempferol and quercetin disrupts quorum sensing and biofilm formation in bacterial communities. Schultz *et al.* [70] found higher relative abundances of Proteobacteria in a quercetin-treated soil compared with non-treated soil. Moreover, alpha diversity indices were observed to significant decline after quercetin treatment [70]. Yu *et al.*, [71] showed that flavones lead to enrichment of the plant beneficial Oxalobacteraceae in the rhizosphere of maize. Guenoune *et al.* [72] reported antifungal effects of the FLV maackiain against the fungal pathogen *Rhizoctonia solani*. Moreover, the enrichment of some members of the order Pleosporales (genus *Paraphaeosphaeria*) in *tt3* and *tt5*, *Westerdykella* in *tt5* and *tt7_7* or depletion of *Alternaria* in *tt7_7*, suggests differential effects of FLVs on members of this order. The species *Holtermanniella takashimae*, which was enriched in *tt3* and *tt5*, was reported to be negatively co-occurring with *Fusarium* species that were pathogenic in wheat [73].

Although *tt5* (impaired in naringenin chalcone) did not affect microbial community composition significantly, naringenin chalcone inhibits spore germination of plant pathogens [68]. In addition, Vandeputte *et al.* [68, 74] demonstrated that plant produced naringenin and catechin is important in reducing the production of quorum sensing-controlled virulence factors in *Pseudomonas aeruginosa* PAO1. The higher number of differentially abundant fungal taxa compared to bacterial taxa, also point to a higher effect of FLVs on fungi, thus supporting the profound role of FLVs on fungi [75].

## Defense signalling mutations have complex effects on microbial taxa

Defense signaling molecules (DSMs) including JA, ABA, SA, FAD and the gaseous molecule ET are well known for their role in mediating plant-microbe interactions. We found that DSM mutants distinctively affected microbial relative abundances and alpha- and beta diversity, thus confirming previous studies [24, 76, 77]. Both *etr1_3* (ET insensitive) and *35S::ERF* (high

                                                                          

ET synthesis) displayed noticeable differences in bacterial and fungal relative abundances and diversity, and thus supports previous studies [78, 79]. The distinct effect of *etr1_3* and *35S::ERF* could be caused by their differential activation of ethylene. The *etr1_3* has reduced ethylene binding activity while *35S::ERF* encodes a transcription factor that regulates plant-microbe interactions, as well as integration of signaling pathways to activate ethylene and jasmonate-dependent responses to pathogens [80, 81]. Using a sterile system with artificially constructed bacterial community, Bodenhausen *et al.* [78] showed that the ethylene-insensitive mutant *ein2* assemble distinct bacterial community compared with the parental line, with a noticeable enrichment of the genus *Variovorax*. Comparably, our study revealed differential enrichment of the genus *Variovorax*, suggesting a selective effect of *etr1_3* on this genus. ABA is an essential molecule in modulating abiotic stress (e.g drought stress and salinity stress), as well as overall plant associated microbial communities [82]. We found that the ABA deficient mutant *aba3_2* affected both bacterial and fungal communities, but surprisingly, only a few bacterial taxa at the genus level, including *Bradyrhizhobium* and *Pseudarthrobacter* were slightly enriched, whereas several fungal genera were enriched. These results indicate a higher antagonistic effect of ABA on fungal communities. In another study, exogenous application of ABA was found to change community composition as well as enrich the genera *Massilia*, *Cellvibrio*, *Limnobacter* [83]. Also, the JA mutant *dde2* (impaired in JA biosynthesis) significantly affected bacterial fungal community composition, with a strong enrichment of bacterial taxa corroborating previous studies [77, 78]. In addition, FAD is pivotal in the phytohormone signalling network by modulating both the SA [84] and JA pathways [85], and its role in mediating plant-microbe interactions has been reported [18, 24]. Likewise, our study revealed distinct effects of FAD mutants on microbial community structures, as both *Azospirillum and Sporobolomyces* were strongly enriched in *fad3_2* and *fad7_1fad8_1*, as was also observed in a previous study in which the Arabidopsis triple mutant *fad3fad7fad8* was enriched in several species within Alpha- and Gammaproteobacteria [24]. Differential effects of FAD genes on microbial taxa have been reported, for instance, while the transcription of the FAD3 gene was shown to be unresponsive upon inoculation of the bacterial pathogen *Xanthomonas campestris* [84], the FAD7 gene was induced by fungal effectors [86]. The distinct enrichment of a number of bacterial genera in the different DSM mutants is indicative of the selective effects of DSMs in shaping the Arabidopsis root microbiome. However, the resident microbial community can interfere with the plant-hormonal pathways [87]. For example, Finkel *et al.* [88] demonstrated that the bacterial genus *Variovorax* utilizes an auxin-degradation operon to alter plant-hormone balances, enabling it to reverse the severe inhibition of root growth that was induced by a wide diversity of bacterial strains. Thus, the analysis of the effect of plant DSMs on microbiomes should be done with caution.

The increasing interest in studying plant metabolites will enable us to better quantify plant host effects on associated microbial communities. However, it is currently challenging to quantify specific effects and mechanisms of important metabolites on plant microbiomes due to methodological limitations. For example, when using mutants, pleiotropic effects arising from gene disruptions in both metabolic and hormonal pathways, makes it impossible to account for individual effects of targeted compounds on microbial community structures. Moreover, because immune signaling activation is complex due to hormonal crosstalk mechanisms it is difficult to quantify the effects of individual hormones on microbial communities. We therefore suggest that, in future studies, detailed analyses should include mutants having complete abolishment of interactive pathways and be complemented with other omics analysis techniques. Furthermore, the host-associated microbiota can alter metabolite synthesis and are also capable of producing several phytohormones [89] and it is therefore important to

adopt experimental approaches that will be able to strictly account for plant-derived compounds and their impact on the plant microbiome.

## Conclusion

Arabidopsis mutants carrying gene disruptions in pathways of the plant secondary metabolites GLS and FLV, or the signalling molecules SA, ABA, ET, or FAD, assembled distinct microbiomes compared to their parental lines. Most earlier studies on the effects of disruption of metabolic pathway have only considered bacterial communities. In this study, we demonstrated dramatic effects of such mutations also on fungal communities. We found distinct relative abundances and diversities of bacterial and fungal taxa in the mutants. Differential analysis at OTU level revealed significantly affected taxa between the mutants and parental lines. Also, the bacterial and fungal genera were mostly enriched than depleted in mutants, except for flavonoid mutations on fungi communities. These results strongly support the perception that many synthesized plant secondary metabolites and DSMs regulate the assembly of the plant root microbiome. However, the interconnectedness in metabolic and signalling pathways presents a high complexity, and thus, mutant lines with several mutations with the possible elimination of overlapping defense-signalling functions are suggested for further studies. The present screening study revealed significant gene-microbiota links, and thus serve as an important resource for in-depth plant-omics analysis in the future.

## Supporting information

**S1 File.**
(PDF)

## Acknowledgments

We thank Profs. Judith Bender (Brown University, USA), Wendy Peer (University of Maryland, USA) and Paul E. Staswich (University of Nebraska, USA) for kindly providing the Arabidopsis mutants. We thank Simone Ena Rasmussen and Mathilde Schiøtt Dige for their excellent laboratory and technical assistance.

## Author Contributions

**Conceptualization:** Enoch Narh Kudjordjie, Rumakanta Sapkota, Mogens Nicolaisen.

**Data curation:** Enoch Narh Kudjordjie, Rumakanta Sapkota.

**Formal analysis:** Enoch Narh Kudjordjie.

**Funding acquisition:** Mogens Nicolaisen.

**Investigation:** Enoch Narh Kudjordjie, Mogens Nicolaisen.

**Methodology:** Enoch Narh Kudjordjie.

**Project administration:** Mogens Nicolaisen.

**Resources:** Mogens Nicolaisen.

**Supervision:** Mogens Nicolaisen.

**Validation:** Rumakanta Sapkota.

**Visualization:** Enoch Narh Kudjordjie.

**Writing – original draft:** Enoch Narh Kudjordjie.

**Writing – review & editing:** Enoch Narh Kudjordjie, Rumakanta Sapkota, Mogens Nicolaisen.

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
