## [Decision Letter · Decision Letter 0]

25 Sep 2021

PONE-D-21-28774Arabidopsis assemble distinct root-associated microbiomes through the synthesis of an array of defense metabolitesPLOS ONE

Dear Dr. Nicolaisen,

Thank you for submitting your manuscript to PLOS ONE. After careful consideration, we feel that it has merit but does not fully meet PLOS ONE’s publication criteria as it currently stands. Therefore, we invite you to submit a revised version of the manuscript that addresses the points raised during the review process. Please submit your revised manuscript by Nov 09 2021 11:59PM. If you will need more time than this to complete your revisions, please reply to this message or contact the journal office at plosone@plos.org. Please include the following items when submitting your revised manuscript:A rebuttal letter that responds to each point raised by the academic editor and reviewer(s). You should upload this letter as a separate file labeled 'Response to Reviewers'.A marked-up copy of your manuscript that highlights changes made to the original version. You should upload this as a separate file labeled 'Revised Manuscript with Track Changes'.An unmarked version of your revised paper without tracked changes. You should upload this as a separate file labeled 'Manuscript'.

We look forward to receiving your revised manuscript.

Kind regards,

Zonghua Wang, Ph.D.

Academic Editor

PLOS ONE

Journal Requirements:

Reviewers' comments:

Reviewer's Responses to Questions

**Comments to the Author**

1. Is the manuscript technically sound, and do the data support the conclusions?

Reviewer #1: Partly

Reviewer #2: Yes

2. Has the statistical analysis been performed appropriately and rigorously? 

Reviewer #1: Yes

Reviewer #2: Yes

3. Have the authors made all data underlying the findings in their manuscript fully available?

Reviewer #1: No

Reviewer #2: Yes

4. Is the manuscript presented in an intelligible fashion and written in standard English?

Reviewer #1: Yes

Reviewer #2: Yes

5. Review Comments to the Author

Reviewer #1: The authors carried out a microbiota survey around the rhizosphere of the selected Arabidopsis mutants that are defects in metabolites such as plant hormone, Glucosinolates, and found that there is a relationship between specific mutations and certain bacterial or fungal groups that are differentially enriched or depleted.

Although there is no mechanistic relationship investigated, and it is hard to conclude if the target metabolites are causative of the microbiota phenotype, this report provide valuable information to generate hypothesis regarding plant metabolites influencing rhizosphere microbiota.

Some suggestions to further improve this manuscript:

1. Please show the identification of the mutants used in this study and if corresponding metoblites are indeed altered in the rhizosphere.

2. For certain metabolites, exogenous application and microbiota survey have been reported. More comparisons between this study and corresponding previous work will provide stronger evidence that the metabolites studied contribute to the phenotype.

3. According to the methods described, the seeds are directly germinated in soil without sterilization. How to make sure if pre-existing microbes in seeds didn’t significantly affects the analysis.

Another issue is about sampling. How to control amount of fine soil attached to roots that are sampled? This will affect how to define rhizosphere in this study.

4. The figures and charts presented are not clear and sharp and should be in higher resolution.

5. Please double-check the references. There was duplicated references (for example, Ref#19, #22).

6. There are some writting errors to be corrected:

Line 71: “single and few mutants” should be “signle and high-order mutants”

Line 83: “Clean” could be deleted

Line 87. “aba3-2” should be italic. Please check other mutant name.

Line196: the nahG mutant expresses a foreign gene of which the product degrades SA.

Reviewer #2: The study entitled “Arabidopsis assemble distinct root-associated microbiomes through the synthesis of an array of defense metabolites” by Kudjordjie et al. took advantage of various Arabidopsis mutant lines defective in the synthesis of glucosinolates (GLS), flavonoids (FLV) and defense hormones (DSMs) to assess their effects on root microbiome assembly. This study revealed dramatic yet complex effects the disruptions of different genes in these pathways have on both bacterial and fungal communities. Overall, this study presents an interesting link between plant genes and root-associated microbiota, the information of which could be further used for a more thorough multi-omics analysis in the future.

Following are my comments:

1. Line 82: 21 mutant lines in total were used in this study. However, in Figure 1, only results for 17 mutant lines were presented. How about the remaining lines? Related to this question, in Figure S2 (A and B), sample npr1_2 showed extremely low numbers of reads and OTUs. What is the reason for that?

2. As the authors have pointed out in the paper, some resident root microbes could potentially degrade or transform plant metabolites. In addition, many root-associated microbes possess the ability to promote plant growth by producing plant hormones or antibiotics to ward off other microbes. Therefore, these possibilities could perhaps interfere with the assembly of root microbiota. In the future experiments, quantitative measurements of root exudates of these mutant lines are warranted to establish the link between plant metabolite and microbiota. Perhaps, a better experimental design is desired in the future to tease out the impacts from plant-derived metabolites and microbe-derived metabolites.

Also some minor corrections need to be made in the text of manuscript:

-Line 158, “Table ST2”: revised to “Table S2”

-Lines 167-168, 170-171: Statements should be accompanied with results of statistical tests.

-Line 186: tt3, tt3_1tt5_1 and tt5 should be compared to their parental line Ler-0, not Col-0.

-Line 203, “p>0.01”: Did the authors actually mean “p<0.01”?

-Line 223, “Bacteroidia”: revised to “Bacteroidetes”

-Lines 238-239: Cite Table S11 in the text.

-Line 259, “iGLS”: Provide the full name.

-Line 266, “…although significant, effects…”

-Line 324, “Holtermanniella takashimae”: This is a species name, not a genus name.

-Line 372: Add “on” before “its host associated microbial communities”.

6. PLOS authors have the option to publish the peer review history of their article (what does this mean?). If published, this will include your full peer review and any attached files.

Reviewer #1: No

Reviewer #2: No

---

## [Author Response · Author response to Decision Letter 0]

11 Oct 2021

Reviewers' comments:

Reviewer's Responses to Questions

Comments to the Author

1. Is the manuscript technically sound, and do the data support the conclusions?

Reviewer #1: Partly

Reviewer #2: Yes

2. Has the statistical analysis been performed appropriately and rigorously? 

Reviewer #1: Yes

Reviewer #2: Yes

3. Have the authors made all data underlying the findings in their manuscript fully available?

Reviewer #1: No

Response

We have provided all data used in this study as stated in the availability of data and materials section “The metadata and sequencing data from MiSeq paired end reads for bacterial 16 s rRNA gene (V3-V4) and fungal ITS2 regions have been deposited in the NCBI Sequence Read Archive (SRA) database under the accession number PRJNA579829” in lines 424-426.

Reviewer #2: Yes

4. Is the manuscript presented in an intelligible fashion and written in standard English?

Reviewer #1: Yes

Reviewer #2: Yes

5. Review Comments to the Author

Reviewer #1: The authors carried out a microbiota survey around the rhizosphere of the selected Arabidopsis mutants that are defects in metabolites such as plant hormone, Glucosinolates, and found that there is a relationship between specific mutations and certain bacterial or fungal groups that are differentially enriched or depleted.

Although there is no mechanistic relationship investigated, and it is hard to conclude if the target metabolites are causative of the microbiota phenotype, this report provide valuable information to generate hypothesis regarding plant metabolites influencing rhizosphere microbiota.

Some suggestions to further improve this manuscript:

1. Please show the identification of the mutants used in this study and if corresponding metoblites are indeed altered in the rhizosphere.

Response

We have provided a detailed description of mutant lines, as well as essential references in Table S1. These lines are well characterized in other studies and constituted the basis for their selection for this study. ¬

2. For certain metabolites, exogenous application and microbiota survey have been reported. More comparisons between this study and corresponding previous work will provide stronger evidence that the metabolites studied contribute to the phenotype.

Response

We acknowledged the suggestion by the reviewer for a more detailed comparisons of our results with previous in-vitro studies. We provided further comparisons as suggested by the reviewer in the revised manuscript. Please, see lines 290-294 and 331-334.

3. According to the methods described, the seeds are directly germinated in soil without sterilization. How to make sure if pre-existing microbes in seeds didn’t significantly affects the analysis.

Response

All the seeds were multiplied in arasystem using the same soil. (https://www.arasystem.com). The aim of the present work was to study microbiomes in a natural setting, and we therefore used non-sterilized seeds (both mutants and their parental lines) that were produced and stored in clean tubes. 

We assume that both mutants and their parental lines would assemble similar microbiomes and in effect would not differentially affect mutant and wildtype Arabidopsis root microbiomes. Moreover, sampling was done 4 weeks after planting, a time period we considered relatively long to enable the manifestation of soil/root-mediated (caused by metabolites) microbiome assembly. 

Another issue is about sampling. How to control amount of fine soil attached to roots that are sampled? This will affect how to define rhizosphere in this study.

Response

To our knowledge, no method for sampling roots/rhizosphere would be completely unbiased. We shook roots until all loose soils were removed. The remaining fine soil and root were then collected for analysis. Although this approach may result in some biases, we were meticulous in ensuring uniformity during shaking. Moreover, our aim was to profile microbiotas associated with the plant compartment (i.e roots and rhizosphere) that are strongly influenced by the metabolites, and thus make this sampling approach more appropriate. 

4. The figures and charts presented are not clear and sharp and should be in higher resolution.

Response

We have regenerated most of the figures (Figs 1-4) and modified Figs-5 and 6 to meet the publication criteria. 

5. Please double-check the references. There was duplicated references (for example, Ref#19, #22).

Response

We have cross-checked the references and edited accordingly. 

6. There are some writting errors to be corrected:

Line 71: “single and few mutants” should be “signle and high-order mutants”

Response

We have rephrased to clarify. See line 72

Line 83: “Clean” could be deleted

Response

This was corrected. See line 87

Line 87. “aba3-2” should be italic. Please check other mutant name.

Response

We corrected and italized all mutants in the text. See line 89.

Line196: the nahG mutant expresses a foreign gene of which the product degrades SA.

Response

For simplicity, we stated that SA is degraded as well in the pen3_1_NahG mutant. See lines 201-205.

Reviewer #2: The study entitled “Arabidopsis assemble distinct root-associated microbiomes through the synthesis of an array of defense metabolites” by Kudjordjie et al. took advantage of various Arabidopsis mutant lines defective in the synthesis of glucosinolates (GLS), flavonoids (FLV) and defense hormones (DSMs) to assess their effects on root microbiome assembly. This study revealed dramatic yet complex effects the disruptions of different genes in these pathways have on both bacterial and fungal communities. Overall, this study presents an interesting link between plant genes and root-associated microbiota, the information of which could be further used for a more thorough multi-omics analysis in the future.

Following are my comments:

1. Line 82: 21 mutant lines in total were used in this study. However, in Figure 1, only results for 17 mutant lines were presented. How about the remaining lines? Related to this question, in Figure S2 (A and B), sample npr1_2 showed extremely low numbers of reads and OTUs. What is the reason for that?

Response

That is correct. 21 mutant lines were used in the study. However, few of the mutant lines including for example, the cyp79B2, pad2-1, fad3_2 npr1_2 in the bacterial dataset had less than 3 replicates after quality filtering (Figure S2) and were thus removed in downstream analysis. We stated this in lines 131-134.

2. As the authors have pointed out in the paper, some resident root microbes could potentially degrade or transform plant metabolites. In addition, many root-associated microbes possess the ability to promote plant growth by producing plant hormones or antibiotics to ward off other microbes. Therefore, these possibilities could perhaps interfere with the assembly of root microbiota. In the future experiments, quantitative measurements of root exudates of these mutant lines are warranted to establish the link between plant metabolite and microbiota. Perhaps, a better experimental design is desired in the future to tease out the impacts from plant-derived metabolites and microbe-derived metabolites.

Response

We agree with this assertion and we strongly anticipated that future studies will be guided by the findings presented herein, in designing inter-omics experiments that will increase our understanding of mechanisms involved in plant-microbiome assembly. 

Also some minor corrections need to be made in the text of manuscript:

-Line 158, “Table ST2”: revised to “Table S2”

Response

Corrected. See line 163.

-Lines 167-168, 170-171: Statements should be accompanied with results of statistical tests.

Response

We provided detailed statistics with asterisks in Figure 1, and text description of Figure 1; “Test of statistical significance was performed by comparing each mutant to their respective background parental lines, and the significantly affected taxa are indicated as *** (P<0.001), ** (P<0.01) and * (P<0.05). ANOVA tests were followed by the Tukey–Kramer post hoc test using the Benjamini and Hochberg (BH) FDR for multiple comparisons”.

-Line 186: tt3, tt3_1tt5_1 and tt5 should be compared to their parental line Ler-0, not Col-0.

Response

Thank you. This was a typo. We corrected it in the revised manuscript. See line 191.

-Line 203, “p>0.01”: Did the authors actually mean “p<0.01”?

Response

Thank you. This was corrected. See line 208.

-Line 223, “Bacteroidia”: revised to “Bacteroidetes”

Response

Thank you. This has been corrected. See line 228

-Lines 238-239: Cite Table S11 in the text.

Response

Thank you. This has been corrected. See line 249

-Line 259, “iGLS”: Provide the full name.

Response

Thank you. This has been edited. See line 264

-Line 266, “…although significant, effects…”

Response

This statement was corrected to … “but significant effect on the microbiome”. See line 272

-Line 324, “Holtermanniella takashimae”: This is a species name, not a genus name.

Response

Thank you. This has been edited. See line 338

-Line 372: Add “on” before “its host associated microbial communities”.

Response

Thank you. This has been edited. See line 387.

---

## [Editor Report · Decision Letter 1]

14 Oct 2021

Arabidopsis assemble distinct root-associated microbiomes through the synthesis of an array of defense metabolites

PONE-D-21-28774R1

Dear Dr. Mogens Nicolaisen,

We’re pleased to inform you that your manuscript has been judged scientifically suitable for publication and will be formally accepted for publication once it meets all outstanding technical requirements.

Kind regards,

Zonghua Wang, Ph.D.

Academic Editor

PLOS ONE
---

## [Editor Report · Acceptance letter]

18 Oct 2021

PONE-D-21-28774R1 

Arabidopsis assemble distinct root-associated microbiomes through the synthesis of an array of defense metabolites 

Dear Dr. Nicolaisen:

I'm pleased to inform you that your manuscript has been deemed suitable for publication in PLOS ONE. Congratulations! Your manuscript is now with our production department. 

Kind regards, 

on behalf of

Prof. Zonghua Wang 

Academic Editor

PLOS ONE